# The impact of the COVID-19 pandemic for children in custody: An analysis of inspection reports

Charlotte Lennox[1]*, Martha Duncan-Zaleski[1], Sahara Nasim[1], Olivia Shirley[1], Kenny Ross[2], Prathiba Chitsabesan[3,4], Louise Robinson[1,5], Jenny Shaw[1], Sarah Leonard[1]

1 Faculty of Biology, Medicine and Health, Division of Psychology and Mental Health, School of Health Science, The University of Manchester, Manchester, United Kingdom, 2 Greater Manchester Mental Health NHS Foundation Trust, Manchester, United Kingdom, 3 Pennine Care NHS Foundation Trust, Manchester, United Kingdom, 4 Manchester Metropolitan University, Manchester, United Kingdom, 5 Lancashire and South Cumbria NHS Foundation Trust, Preston, United Kingdom

* charlotte.lennox@manchester.ac.uk

## Abstract

### Background

Children in custodial settings are a vulnerable group. Prior to the COVID-19 pandemic there were concerns about the safety of children in these settings. COVID-19 has had an impact on everyone but given the vulnerability of children in custody, there were concerns about the impact of COVID-19 restrictions. All custody settings for children are independently inspected and this research aimed to analyse data from inspection reports. Twenty-six inspection reports undertaken between March 2020 and October 2021 were analysed to understand the impact of COVID-19 on delivery of usual care/regime.

### Results

Data showed that across all site's children spent considerable amounts of time isolated and in some cases, this was deemed to amount to solitary confinement. There was evidence of some positive experiences, in the smaller sites, around COVID-19 slowing the pace of life allowing staff and children could foster relationships. However, in the larger sites, isolation was extreme and COVID-19 policies such as 'bubbles' appear to have created unintended consequences as sites have moved into recovery, leading to increased violence and stress. COVID-19 directly impacted staffing levels. This and the COVID-19 policies to reduce mixing also had an impact on how children's behaviour, welfare and safeguarding was managed. In some larger sites, being COVID-19 secure was prioritised over the needs of the children.

### Conclusions

This research highlights the importance of multi-site longitudinal research to understand how children, staff and institution's function. The experiences of children in custody during

**Data Availability Statement:** All relevant data are within the manuscript and its Supporting Information files.

**Funding:** This project is funded by the National Institute for Health Research (NIHR) Policy Research Programme [NIHR202660]. The views expressed are those of the author(s) and not necessarily those of the NIHR or the Department of Health and Social Care. The funders had no role in study design, data collection and analysis, decision to publish, or preparation of the manuscript.

**Competing interests:** The authors have declared that no competing interests exist.

**Abbreviations:** HMIP, His Majesty's Inspectorate of Prisons; HMPPS, His Majesty's Prison and Probation Service; Ofsted, The Office for Standards in Education; SCH, Secure Children's Home; STC, Secure Training Centres; YOI, Young Offender Institution.

COVID-19 differed by site type. The research suggests that the larger sites are struggling to keep children safe and there should be a shift towards smaller, more therapeutic environments. More research is needed to understand the longer-term unintended consequences of COVID-19 policy in custody, for these vulnerable children.

## Introduction

Children who are remanded or sentenced to custody in England and Wales are accommodated within three distinct types of secure setting. These are Young Offender Institutions (YOIs), Secure Training Centres (STCs), and Secure Children's Homes (SCHs). These vary in size, function, and operator. YOIs are the largest settings and accommodate boys aged 15–17. Since March 2022, one YOI has also accommodated girls due to the closure of one of the STCs. As of December 2022, there were 377 children in YOIs [1]. STCs are mid-sized settings and typically accommodate boys and girls aged 12 to 17. As of December 2022, there were 46 children in STCs. While the YOIs and STCs come under the prison service, the SCHs operate within the social care system, are much smaller in size (the largest having 38 beds) and accommodate children from the age of 10–17, with the highest staff ratio. As of December 2022, there were 68 children in SCHs.

Prior to the COVID-19 pandemic, concerns had been raised about the safety of children in custodial secure settings in England and Wales, with reports stating that no YOI or STC was safe [2,3]. Self-harm rates had increased by 35% and the use of restrictive physical interventions (situations where force is used to overpower or with the intention of overpowering a child) had increased by 19% [4]. On 23rd March 2020, the UK went into 'lockdown' as a response to the COVID-19 pandemic. Restrictions and limited social interaction had an impact on the wellbeing of the general population [5] and a disproportionate impact on adolescents' [6].

As there were already concerns prior to COVID-19 about the conditions for children in custody there were fears about the impact of lockdown [7,8]. In response to the COVID-19 pandemic secure settings adopted a limited and restrictive regime. The implementation of physical distancing, infection control, and hygiene measures caused disruption to the children's daily routines. Family visits were halted, as well as education/work opportunities, and access to offending behaviour and most mental health interventions. In some settings, children were isolated in their rooms for nearly 24 hours a day, with only 15–30 minutes being allocated for a shower or fresh air [9]. In the early days of the pandemic the impact of these measures was not known. However, in a scoping review [10] of 83 articles of the effects of the COVID-19 pandemic for adult prisoners, the main anxieties voiced by those detained were regarding a lack of meaningful social interactions; an increased use of solitary confinement; increased stress and anxiety; disruption to family and friends' visits; and health anxiety about loved ones.

In 2022, the prison service published research on the impact of the COVID-19 pandemic on youth custody [11]. Data were collected from a range of stakeholders including children, custody staff, parents, and senior leaders, from July to November 2020. The report concluded with four strategic recommendations: that being in smaller 'family' groups could help children feel safer, particularly in YOIs; that the lockdown experience had allowed staff and children to develop more positive relationships and attempts should be made to foster this relationship going forward; that children need the opportunity, space and support to develop formal but more importantly informal learning/skills; and that strategic decisions made by the prison

service did not allow for the specific needs and best interests of children to be prioritised. While this is valuable research, it only covered experiences during a small period of the COVID-19 pandemic, and it is important to understand longer term impacts of the restrictions.

All custody settings for children are independently inspected on a regular basis to provide scrutiny of the conditions and services provided. Researchers have often neglected documentary analysis [12], yet this form of investigation allows for nuanced understanding of how people are supported and how services are delivered within the context of secure environments [13–15]. It also allows for an evaluation of the longitudinal and collective experiences, which cannot be obtained from individual inspection reports alone [16]. This paper aims to analyse data from inspection reports for the secure settings for children in custody in England and Wales to understand the impact of the measures implemented during the COVID-19 pandemic.

## Materials and methods

### Design

A qualitative content analysis was conducted, analysing inspection reports on the delivery of secure settings for children in custody in England and Wales. This was a desk-based study of data in the public domain.

### Ethics

The study protocol was reviewed by The Division of Psychology and Mental Health Ethics Committee and deemed not to require other types of ethical approval.

### Sample

We accessed 26 inspection reports across all 14 children's secure sites (5 x YOIs, 2 x STC, 7 SHCs) undertaken between March 2020 and October 2021. The inspection reports were obtained between October and November 2021. In total there were 12 full inspection reports (full inspections are unannounced and cover all areas) for all but three settings. In two SCHs, only short interim inspections were available and in one YOI there was only a combined inspection with another YOI available (see Table 1). Some sites received more than one inspection during our window of data collection. Links to all the inspection reports analysed in this study are available in S1 File.

### Data extraction

Each of the three setting types are inspected differently. The YOIs are inspected by His Majesty's Inspectorate of prisons (HMIP) and follow the healthy prison test [17]. This includes a focus on safety, care, purposeful activity and resettlement. The STC's are inspected under a joint inspection [18] assessing overall experiences and progress of children, including judgements on children's education and learning; children's health; children's resettlement; taking into account how well children are helped and protected; and the effectiveness of leaders and managers. The SCHs are inspected by The Office for Standards in Education (Ofsted) under a Social Care Common Framework [19], to determine overall experiences and progress; children's education and learning; children's health; and how well children are helped and protected; and how well the home is fulfilling its expectations.

**Table 1. During COVID-19 reports' site type and inspection type and date.**

| Site ID | Site Type | Inspection type | Inspection Date |
|---------|-----------|-----------------|-----------------|
| YOI 1 | YOI | Combined & Full | Apr-20 & Aug-21 |
| YOI 2 | YOI | Combined & Full | Jul-20 & Feb-21 |
| YOI 3 | YOI | Combined & Full | Apr-20 & Apr-21 |
| YOI 4 | YOI | Combined | Jul-20 |
| YOI 5 | YOI | Combined & Full | Apr-20 & Jan-21 |
| STC 1 | STC | Short x 3 & Full | Nov-20, May-21, Sept-21 & Oct-21 |
| STC 2 | STC | Short x 3 & Full | Oct-20, Dec-20, Jan-21, Jun-21 |
| SCH 1 | SCH | Full | May-21 |
| SCH 2 | SCH | Short x 2 | Mar-20 & Nov-20 |
| SCH 3 | SCH | Short & Full | Nov-20 & Apr-21 |
| SCH 4 | SCH | Short | Aug-21 |
| SCH 5 | SCH | Short & Full | Sept-20 & Jun-21 |
| SCH 6 | SCH | Full | May-21 |
| SCH 7 | SCH | Short & Full x 2 | Sept-20, Apr-21 & Jun-21 |

To ensure data were extracted from each inspection report in a comparable manner, a data extraction framework was developed to provide consistency across all the different inspections, which corresponded to the format of HMIP inspection reports (see Table 2).

## Procedure

The inspection reports were accessed from the websites hosting the public repository e.g., HMIP & Ofsted. We also accessed the last full inspection report for each site pre-COVID-19: these inspections ranged from November 2018 and February 2020 (see Table 3). These were read in full and attributed a rating of red, amber or green in order to distinguish the rating of performance for the setting. This was based on the ratings used by the inspectors within the reports. Green was outstanding/good, amber was good/reasonably good and red was requires improvement/inadequate/not sufficiently good/poor. We implemented this to ensure that the inspection reports from during-COVID-19 were read within the context of how each site was performing previously, when ascribing the impact of COVID-19 (see S1 File).

## Analysis

The during-COVID-19 reports were entered into NVivo [20], categorised based on pre-COVID-19 performance and data were extracted based on the domains and sub-domains

**Table 2. The four inspection domains and corresponding sub-domains.**

| Keeping children safe | Care | Education, learning and activity | Children's Journey |
|-----------------------|------|----------------------------------|--------------------|
| Early days | Daily life | Amount of activity | Family contact |
| Safeguarding | Relationships | Education/training/skills | Planning for the future |
| Behaviour | Health | | Education planning |
| Safety | Substance misuse | | Transition |
| Self-harm and suicide | Equality, diversity & inclusion | | Interventions/support |
| Bullying and violence | | | |
| Restrictive practices | | | |
| Separation | | | |

**Table 3. Pre COVID-19 reports' site type and inspection type and date.**

| Site ID | Site Type | Inspection type | Inspection Date |
|---|---|---|---|
| YOI 1 | YOI | Full | Sep-19 |
| YOI 2 | YOI | Full | Jul-19 |
| YOI 3 | YOI | Full | Nov-19 |
| YOI 4 | YOI | Full | Jan-20 |
| YOI 5 | YOI | Full | Mar-19 |
| STC 1 | STC | Full | Apr-19 |
| STC 2 | STC | Full | Nov-18 |
| SCH 1 | SCH | Full | Jan-20 |
| SCH 2 | SCH | Full | Dec-19 |
| SCH 3 | SCH | Full | Feb-20 |
| SCH 4 | SCH | Full | Jan-20 |
| SCH 5 | SCH | Full | Oct-19 |
| SCH 6 | SCH | Full | Feb-20 |
| SCH 7 | SCH | Full | Jun-19 |

highlighted in Table 2. The extracted data were then inductively coded in order to identify emerging patterns and themes around the impact of COVID-19 restrictions and practices.

## Findings

Upon analysing the inspection reports, it became clear that there were key ways in which COVID-19 and the restrictions impacted children in custody, these included; isolation, the use of 'bubbles', staffing shortages and the subsequent impact of the staffing shortages on behaviour management, welfare and safeguarding and the delivery of interventions to reduce offending.

## Isolation

Children isolating for up to 14 days on arrival was a measure used in all sites. In the YOIs and STCs, this was a policy known as Reverse Cohorting Unit (RCU) [21]. Across inspection reports there was variation in how much time children were allowed to spend out of their rooms, while in the RCU. In one YOI, children received 4.5 hours face-to-face education a day, half an hour in the fresh air and an additional 90 minutes to take a shower and participate in activities with staff to prevent social isolation (YOI 3, Apr 2021). This was notably higher than the other YOIs and was due to the proactive work of local leaders in recovery planning. In a multi-site report of two YOIs, children expressed frustration to inspectors at being locked in their rooms for up to 23 hours a day, only having 45 minutes in the open air and 15 minutes for a shower. Some children were only given access to showers and exercise twice a week (YOI 4, 2, Jul 2020). In the STCs, children spent about 19 hours per day locked in their room. In one STC this increased to 23 hours (STC 1, Sept 2021) and 23.5 hours in the other STC (STC 2, Dec 2020). Inspections reported that the implementation of RCU practice amounted to solitary confinement (STC 2, Oct 20).

> "Children are locked in their bedrooms all day and night, only being allowed out for 30 minutes each day. Records from these 14-day periods [in the RCU] show that there is no meaningful interaction with children. This is an excessive amount of time for children to be

locked in, is tantamount to solitary confinement, and is highly likely to be damaging to their emotional and physical well-being" (STC 2, Oct 2020)

The isolation period in the RCU was described as being used positively in many sites, with staff taking this opportunity for 1:1 support and to complete assessments on the children. This allowed them to identify any additional needs; adjust their living or health plan; and gather information from parents and professionals. In one YOI, staff began to build positive relationships with the children (YOI 3, Apr 2021), however, in a multi-site inspection report of three YOIs it was reported that:

"Because these children were not well-known by staff, they were invariably less well-supported in managing their behaviour and emotions. While the restrictive regimes were necessary for keeping all children safe, they clearly had an impact on the support that could be offered to children who were new to custody and their access to meaningful human interaction" (YOI 1, 3, 5, April 2020)

Once admission isolation was completed, general time out of room also varied across sites and during the different waves of the COVID-19 pandemic.

As restrictions lessened, children had access to a more relaxed regime in some YOIs. By 2021, most sites gave children about 4.5 hours a day out of their room, except for one site which allowed 10 hours on weekdays (YOI 3, Apr 2021). However, despite the easing of government restrictions during 2021, in some YOIs some children continued to spend less time out of their rooms—this tended to be children on specialist units for vulnerability or risk (YOI 1, Aug 2021). The inspections reported little information on SCHs approach to isolation. The inspection reports, in general, found that this limited time out of room did have an impact on the children:

"As at the time of the previous inspection, children spent too much time locked in their cells. This affected their relationships with staff because there was little time to build trust and respect. In addition, there was no system to make sure that children had regular, meaningful contact with a named officer, and most residential staff we spoke to were unaware of how the children in their care were progressing in education or other areas of life at the establishment." (YOI 1, Aug 2021)

## Bubbles

The use of 'Bubbles' was a strategy implemented across all sites. Bubbles were a process of creating a small group or 'household'. This meant that people could have close contact with others in that household bubble and did not need to maintain social distancing practices. Three types of bubbles were observed by inspectors across the YOIs and STCs: 1) 'Post admission/isolation bubbles': created when groups of children arrived together and isolated in the RCU or when groups of children tested positive for COVID-19; 2) 'Family' bubbles: small groups of children located on the main living locations; and 3) 'Education' bubbles: groups that completed face-to-face education or activities together which, in some sites, was the same group of children housed in living bubbles. The SCH inspection reports infrequently mentioned the use of bubbles, mainly due to the small 'family' units' children were located in pre-pandemic, leading to restrictions only limiting cross-unit mixing.

The inspection reports highlighted that the use of post admission/isolation and family bubbles appear to have resulted in violence. An inspection report from one STC suggested that

grouping based on COVID-19 status and not on the needs/risks associated with the children was an issue.

> "Violence and self-harm within the centre had increased over the three months prior to this visit. The centre's attempts to manage the safety of children during the COVID-19 outbreak, by separating children into 'bubbles' of those who had tested positive and those who had tested negative for COVID-19, resulted in issues with children's compatibility and an increase in both peer-on-peer violence and assaults on staff" (STC 2, Jan 2021)

In terms of family bubbles, initially these were successful as the smaller groups were easier for staff to manage behaviour; maintain good communication; and trusting relationships between staff and children. However, the bubbles led to rising tensions and inter-bubble conflict after months of restrictions. Conflict within bubbles was reported as common and the management of inter-bubble conflict was reported as unsatisfactory by inspectors. Response to intra-bubble conflict usually involved making bubbles smaller, exacerbating the existing tension and resulting in further violence. This tension continued as children reported the need to carry a weapon to feel safe. In some sites, staff use of force remained high or increased in response to the violence caused by housing bubble conflicts and was also felt to be a factor causing increase in self-harm rates (STC 1, May 2021; YOI 2, Feb 2021; SCH 2, Nov 2020).

Within the larger sites, as they moved into recovery the need to expand the size of bubbles for the regime to operate, became a flash point.

> "Of concern was the frequency of multiple perpetrator assaults, where two or more children would attack a lone child simultaneously. This had been identified in the safety meeting as a major risk, especially as the prison moved toward full landing 'communities' rather than the current small groups of between one and 10 children. Throughout the residential units, staff strictly controlled the unlocking of any cell door while children from a different sub-group were on the landing. This reflected a lack of staff confidence in managing individual children, and the widespread belief that children would attack each other at any opportunity" (YOI 1, Aug 2021).

Education bubbles were introduced later into the pandemic, when education could return to in-person teaching at various points during 2021. Education bubbles would consist of 2–4 children and tended to have the same children as family bubbles, with teaching adapted to the differing educational levels. Education bubbling was seen as successful in most sites. In comparison to pre-pandemic, education in much smaller groups, despite having mixed-abilities, aided learning by making children feel safer and less subject to peer pressure, resulting in increased engagement with their education. Behaviour in education settings improved, as lessons became calmer environments, and attendance was also high.

## Custody staff shortages

COVID-19 resulted in custody staffing shortages and this impacted almost every aspect of site functioning. Staff levels were depleting throughout the pandemic due to health risks, shielding, isolating, and illness. While 'safe' staffing levels were maintained in most sites, apart from one STC that was not meeting the contractual minimum requirements on several occasions, there was a reduction in what procedures and activities could be safely carried out in the sites (STC 1, May 2021)

Staffing levels were reported to be one (of many) factors related to adverse events such as violence, bullying, self-harm and challenging behaviour, and across sites in general, staff

confidence to manage volatile situations was impacted. Violence towards peers and staff increased across most sites despite a reduction in the number of children detained. Children also reported verbal abuse by staff, with one YOI survey demonstrating that a third of children across the site reported having been verbally abused by staff (YOI 5, Jan 2021). Poor staffing levels and limited supervision of children allowed for children to bully and intimidate each other. In one STC inspection report behaviour such as play fighting and swearing often went unchallenged due to the limited staffing (STC 2, Jun 2021). Physical assaults on peers were common during the pandemic and minimal conflict resolution by the small number of staff aided this rise in assaults in YOIs. Analysis of adjudications had been stopped during the pandemic with no quality assurance in place, as found in one YOI inspection report, as such the adjudications process was undermined, and children suffered no consequences for their poor behaviour.

## Behaviour management

Behaviour management schemes were relied upon during the pandemic to control children's behaviour. All sites reported having a form of behaviour management scheme, allowing children to earn or lose privileges. Behaviour management schemes at STCs were deemed by inspectors as not effective at managing children's behaviour (STC 2, Oct, 2020).

> "Neither staff nor children feel that the rewards and sanctions included in this new system promote good behaviour. For example, some staff said that they felt their colleagues would undermine them by not enforcing the sanctions and so were reluctant to give them, and the green cards which are given out for rewards are not readily available. Children said that they could do what they want without sanction and the rewards did not encourage them to behave well." (STC 2, Oct 2020).

Behaviour management systems or incentive schemes were level-based in most sites (e.g., bronze, silver, gold). Several YOIs made changes to the lowest level, for example ceasing the removal of televisions as a punishment during the pandemic (YOI 1, Aug 2021; YOI 2, Feb 2021). However, as the lower and higher levels of the behaviour management system became less distinct and all children were following the same limited regime, children became demotivated.

> "The behaviour management policy contained three main elements: an instant reward and sanction scheme; an incentives scheme with three different levels; and an adjudication system, which dealt with the most serious incidents of poor behaviour. The governor's vision was that this would be underpinned by consistent staffing: this would allow them to build relationships with children, provide a stable environment and motivate them to engage. In practice, however, staff were frequently cross deployed and there were few meaningful incentives for those who behaved well or engaged in education or rehabilitation activities. Instead, staff relied on keeping children apart in small groups to manage conflict. This and the poor provision of activities and time unlocked meant that there was little to motivate children to use their sentence usefully" (YOI 1, Aug 2021).

Across the YOIs and STCs, where incentive schemes did not manage behaviour, staff would often then try to resolve issues by moving children: the most common response was to move children to a different bubble. However, as highlighted, use of bubbles, led to many difficulties. Across the YOIs and STCs where moving children to different bubbles proved ineffective, staff resorted to the use of force and restraints in response to behaviour. STCs saw an increase in

levels of force used against children (STC 1, May 2021). Pain-inducing restraints were also relied on in YOIs, with one child reported to receive three pain-inducing techniques during one incident (YOI 2, Feb 2021). In another YOI there were unauthorised punishments being used including turning off a child's water in their room (YOI 5, Jan 2021). Children in one STC expressed to inspectors that the unsafe environment created by physical restraints, violence, verbal abuse, and poor behaviour left unchallenged meant there was an increase in the making and use of improvised weapons. Children reported that they have weapons 'just in case'.

At SCHs, inspection reports noted that children were contributing to their behaviour management plans and staff had adapted routines to meet the children's needs, to reduce conflict or flash points. Good relationships between children and staff and the limited use of force and restraint fostered this cooperation over behaviour management. Overall behaviour management strategies were effective in SCHs over the pandemic and the use of sanctions decreased in one SCH due to staff promoting positive behaviour effectively (SCH 3, Apr 2021).

Within the inspection reports separation was used to manage poor behaviour, especially in response to low staff numbers. There was a particular increase in single separation in STCs as a way of managing behaviour (STC 1, Sep 2021). In most YOIs there was a reduction in the use of separation, however in one YOI a child had not left their room in four days (YOI 1, Aug 2021). SCHs, on the other hand, had good scrutiny and oversight of single separation to ensure it was used according to policy to manage behaviour or COVID-19.

## Welfare and safeguarding

Throughout the pandemic, staff were relied upon to complete welfare checks on children and make decisions regarding safeguarding due to the sudden lack of family contact and the restrictions that came with the COVID-19 pandemic. In YOIs, there was an increase in welfare checks and most claimed to carry them out every day, with staff making appropriate referrals to the relevant safeguarding teams. However, in one YOI staff were unaware of how to carry out welfare checks, putting children at risk (YOI 2, Feb 2021). At one STC it was reported that welfare checks were not followed up, undermining their purpose (STC 1, Oct 2021). In the YOIs and STCs, the process for managing self-harm and suicide risk continued throughout the pandemic however there were reports that the quality of these documents had deteriorated in the pandemic in one YOI (YOI 1, Aug 2021). Sections of care plans were left incomplete including critical sections such as risks, triggers and protective factors, and the reviews rarely generated further actions. In one of the SCH report the inspectors noted an increase in levels of self-harm following the implementation of restrictions, in spite of there being an increase in the staff support for children vulnerable to self-harm in the pandemic. The report stated that the rise in self-harm in one SCH was due to the increased level of separation and physical restraint (SCH 2, Nov 2020).

## Interventions to reduce reoffending

At most sites, all accredited intervention work to address offending behaviour stopped at the beginning of the pandemic, in a few sites this continued as 1:1 work rather than in groups. When interventions restarted, as points during 2021, there were large backlogs of children waiting for interventions and consequently shorter and less in-depth interventions took place (YOI 1, Aug 2021; YOI 5, Jan 2021). Due to this backlog, many children were released without having had their offending behaviour addressed.

## Discussion

The research undertook a content analysis of data extracted from inspection reports for children in custody in England and Wales, to understand the impact of COVID-19 and

restrictions on the delivery of care. Data were analysed from 26 inspection reports across the 14 secure sites, with inspections taking place from March 2020 to October 2021. The analysis showed variability in the levels of isolation experienced by children across sites and that 'bubbles', as a way to minimise the spread of COVID-19, had an unintended consequence of leading to more conflict and violence. COVID-19 had a major impact on staffing levels and in response to these depleted staffing levels changes were made to the delivery of care, which further impacted the children, for example, the way in which behaviour, welfare and offending behaviour were addressed.

An important finding of this research is that across all themes, the extent of the impact was not felt equally across the different site types. The SCHs appear to have been able to run with more flexibility and autonomy to meet the needs of the children, while the YOIs and STCs seem to have prioritised security. Shaun Common, Ofsted's Senior Officer said that "an almost normal routine for children" in SCHs was being kept in "stark contrast" to YOIs [22]. This is attributed to an experienced, well-trained and resilient workforce with skilled leaders and managers. SCHs have a 'child first' culture and with high staff to child ratios, most SCHs are performing well. However, that is not to say that there were no impacts felt by children in SCHs, whilst these were present [23] they were much less extreme than the STCs and YOIs. It would appear from the inspection reports analysed and reports prior to COVID-19 [2,3] that there are serious concerns about SCTs and YOIs ability to provide safe care. However, over the last 20 years, 16 SCHs have closed across England [24], resulting in the majority of children now being accommodated in STCs and YOIs, therefore it would suggest the need for a serious rethink of how the future of secure care should look. England's review of children's social care was published in May 2022. The review did recommend that all YOIs and STCs be closed in the coming years, to be replaced by a network of small secure children's homes to ensure there is sufficient capacity. To support this, 20 Regional Care Cooperatives will be established across England, creating new secure care within their regions [25].

Another important finding relates to the use of 'bubbles', the process of grouping small numbers of children together. Of the three types of bubbles discussed in the inspection reports, education bubbles appear to be the most successful as children felt less anxious and pressured and could engage better in smaller groups. A recent systematic review of factors identified by children with behavioural, emotional and social difficulties as influential on their experiences of education, found that more personalised learning and the adapted curriculum were commonly considered important to positive schooling experiences [26]. Whilst other types of bubbles initially allowed staff and children to foster better relationships, the use of them over the longer term appears to have created separation, division and conflict, and increased stress and anxiety within the children. It is a concern that the COVID-19 report from the prison service [11] recommended the wider implementation of family groups ('bubbles'). It is likely that due to the data collection being a relatively short snapshot early on in the pandemic, any unintended consequences of policies could be not examined. Family units work well in smaller sites and the SCHs operated in this way prior to COVID-19. The issue with 'bubbles' seems to have arisen in the larger sites where 'bubbles' did not mix at all for long periods of time and were not then expanded to operate a functioning regime. This was due to the size of the establishments and staffing levels but resulted in the children not being supported or prepared for this transition. Our research does indicate that a fundamental shift to moving children to smaller units may be more effective, or larger sites with children in smaller self-contained units, akin to a therapeutic community. However, there is also a need for further research to understand how best to mobilise any transitions for children.

In terms of impact of COVID-19 on the delivery of usual care/regime, the inspection reports highlighted impact on behaviour management, safeguarding and meeting children's

needs. But by far the biggest impact shown within the inspection reports was the impact of custody staffing shortages. It must be highlighted that staffing issues were already a concern pre-pandemic [27] but the impact of COVID-19 exacerbated an already serious issue. Prison service data suggests that staff working with children had the highest sickness absence across all prison areas with 10.2 Average Working Days Lost (AWDL) as of end March 2021 [28]. This is in addition to a retention and recruitment issue [29]. Inspection report data limits our ability to unpick the causal impact of COVID-19 over and above that of staff shortages, but it appears to at the very least, have made a bad situation worse. The Framework for Integrated Care (SECURE STAIRS) was rolled out for children in custody from 2018, with the aim to promote consistent, trauma-informed, formulation-driven, evidence-based care, delivered within a whole-systems approach by well trained and supported staff. One of the outcomes of this approach was to improve staff recruitment and staff retention, but as yet it does not seem to be achieving this [30].

## Strengths and limitations

A strength of this study is that it included data across every site holding children in custody ensuring the collective experience was captured. The inspection reports covered a large window of the COVID-19 pandemic, from March 2020 and November 2021, allowing for a longitudinal understanding of the impact, something that another research was unable to provide [11]. A particular strength is in the study methodology: by accessing the pre-COVID-19 inspections we were able to ensure that the during-COVID-19 reports were read within the context of how each site was performing previously, allowing us to establish if good or poor performance was due to COVID-19 restrictions. A limitation of the study, in common with many studies using documentary analysis, is that the inspection reports were not produced for research purposes and therefore detail was lacking in places [31]. Additionally, not all sites had full inspection reports and one site (YOI 4) was only captured within a combined inspection, therefore this research may not reflect that site's full experience. A challenge was to develop a consistent framework for analysis across three site types with their own inspection criteria. A decision was made to use the domains used within the YOI inspection framework as this provided the most domains and sub-domains for coding, but we changed the language and focus to capture the SCH inspection content.

## Conclusions

This research highlights the importance of collective and longitudinal research to understand how children, staff and institution's function and the unintended consequences of policy change over time. There is increasing evidence that children are much more likely to be indirectly, than directly, affected by COVID-19, due to education changes, healthcare delivery changes and wider societal changes [32]. Children in custody are just as, if not more, likely to be indirectly impacted as a result of their existing health and social inequalities [33]. The experiences of children in custody during COVID-19 differed by site type and more research is needed to understand and unpick the indirect longer consequences of COVID-19 policy changes in custody for children. However, children in custody spent considerable time isolated and this will undoubtedly have long term implications. This research supports the growing calls for a major rethink about our approach to children in custody. There is increasing evidence that smaller more therapeutic units are the direction of travel but considering the large numbers of SCHs already closed and repurposed, there is likely a huge financial implication to this change.

## Supporting information

**S1 File.**
(DOCX)

## Author Contributions

**Conceptualization:** Charlotte Lennox, Sarah Leonard.

**Data curation:** Charlotte Lennox.

**Formal analysis:** Charlotte Lennox, Martha Duncan-Zaleski, Sahara Nasim, Olivia Shirley, Sarah Leonard.

**Funding acquisition:** Charlotte Lennox, Kenny Ross, Prathiba Chitsabesan, Louise Robinson, Jenny Shaw.

**Investigation:** Charlotte Lennox, Sarah Leonard.

**Methodology:** Charlotte Lennox, Sarah Leonard.

**Project administration:** Charlotte Lennox.

**Resources:** Charlotte Lennox.

**Supervision:** Charlotte Lennox, Kenny Ross, Prathiba Chitsabesan, Louise Robinson, Jenny Shaw.

**Validation:** Sarah Leonard.

**Visualization:** Sarah Leonard.

**Writing – original draft:** Charlotte Lennox, Martha Duncan-Zaleski, Sahara Nasim, Kenny Ross, Prathiba Chitsabesan, Louise Robinson, Jenny Shaw, Sarah Leonard.

**Writing – review & editing:** Charlotte Lennox, Martha Duncan-Zaleski, Kenny Ross, Prathiba Chitsabesan, Louise Robinson, Jenny Shaw, Sarah Leonard.

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
