## [Decision Letter · Decision Letter 0]

8 Oct 2023

PONE-D-23-20933The impact of the COVID-19 pandemic for children in custody: an analysis of inspection reportsPLOS ONE

Dear Dr. Lennox,

Thank you for submitting your manuscript to PLOS ONE. After careful consideration, we feel that it has merit but does not fully meet PLOS ONE’s publication criteria as it currently stands. Therefore, we invite you to submit a revised version of the manuscript that addresses the points raised during the review process.

We look forward to receiving your revised manuscript.

Kind regards,

Francesco De Micco, M.D., Ph.D.

Academic Editor

PLOS ONE

“This project is funded by the National Institute for Health Research (NIHR) Policy Research Programme [NIHR202660]. The views expressed are those of the author(s) and not necessarily those of the NIHR or the Department of Health and Social Care.”

Reviewers' comments:

Reviewer's Responses to Questions

**Comments to the Author**

1. Is the manuscript technically sound, and do the data support the conclusions?

Reviewer #1: Yes

Reviewer #2: Yes

2. Has the statistical analysis been performed appropriately and rigorously? 

Reviewer #1: N/A

Reviewer #2: N/A

3. Have the authors made all data underlying the findings in their manuscript fully available?

Reviewer #1: Yes

Reviewer #2: No

4. Is the manuscript presented in an intelligible fashion and written in standard English?

Reviewer #1: Yes

Reviewer #2: No

5. Review Comments to the Author

Reviewer #1: This is a straightforward, descriptive paper about the impact of Covid-19 on children in custody in the UK. The paper is well structured and well written albeit quite lengthy. I would recommend it is published and only have a few comments:

• By the time I had read to the end of the lengthy Findings section, I was unsure what the most important point was that the author team wanted to reader to take home. Likewise, the Abstract does not have a central main point pulling it all together. The paper is pitched as being about the mental health of children in custody but there was less emphasis on exploring this and significantly more descriptive content about the practicalities of Covid-19 restrictions and practices in child custodial settings. I think the author team should either a) slightly re-write elements of the paper to turn it into a paper that focuses more on process of Covid-19 restrictions or b) they should write a new section which firmly brings together all the findings to make clear statements about how Covid-19 impacted on children’s mental health.

• I was unsure how the analytic framework was developed (line 206). Were these headings that are already in the inspection reports? If so, how did you allow for inductive findings to arise? If not, could you add in some detail about how the framework was developed?

Reviewer #2: While the findings of this study are of interest it is important that the authors redraft to adddress some major deficits before resubmitting.

The Discussion section is inadequate and contains too much that is in effect a summary of findings and not a discussion of the implications of those findings.

Given that the article would be read by an international audience, a box summarising the agencies with their acronyms and responsibilities would be very useful if not essential.

Similarly some terms such as 'bubbles' took a special meaning during COVID and should be explained.

I would advise the authors to avoid use of significant as the changes etc were not measurable and it introduces potential confusion.

Overall there are too many grammar/ punctuation errors which indicate the need for the article to be corrected before submission.

6. PLOS authors have the option to publish the peer review history of their article (what does this mean?). If published, this will include your full peer review and any attached files.

Reviewer #1: No

Reviewer #2: No

---

## [Author Response · Author response to Decision Letter 0]

21 Dec 2023

Reviewer #1: This is a straightforward, descriptive paper about the impact of Covid-19 on children in custody in the UK. The paper is well structured and well written albeit quite lengthy. I would recommend it is published and only have a few comments:

• By the time I had read to the end of the lengthy Findings section, I was unsure what the most important point was that the author team wanted to reader to take home. Likewise, the Abstract does not have a central main point pulling it all together. The paper is pitched as being about the mental health of children in custody but there was less emphasis on exploring this and significantly more descriptive content about the practicalities of Covid-19 restrictions and practices in child custodial settings. I think the author team should either a) slightly re-write elements of the paper to turn it into a paper that focuses more on process of Covid-19 restrictions or b) they should write a new section which firmly brings together all the findings to make clear statements about how Covid-19 impacted on children’s mental health.

The introduction and elsewhere has been edited to remove the emphasis on mental health and more clearly highlight the focus on the impact of COVID-19 restrictions and practices. The results section has been reduced to focus more clearly on the impact of COVID-19 restrictions and practices. 

• I was unsure how the analytic framework was developed (line 206). Were these headings that are already in the inspection reports? If so, how did you allow for inductive findings to arise? If not, could you add in some detail about how the framework was developed?

We have provided more clarity around the analysis and also tried to remove the confusion around the term framework in relation to the frameworks for the inspections vs. an analytical framework. 

Reviewer #2: While the findings of this study are of interest it is important that the authors redraft to address some major deficits before resubmitting.

The Discussion section is inadequate and contains too much that is in effect a summary of findings and not a discussion of the implications of those findings.

The discussion has been rewritten to reflect the changes to the findings section but also to provide more discussion of the implications.

Given that the article would be read by an international audience, a box summarising the agencies with their acronyms and responsibilities would be very useful if not essential.

There is already a list of acronyms provided within the manuscript, therefore rather than duplicating this we have summarised the responsibilities within the manuscript and reduced the use of agencies for international audiences.

Similarly some terms such as 'bubbles' took a special meaning during COVID and should be explained.

These terms have been explained.

I would advise the authors to avoid use of significant as the changes etc were not measurable and it introduces potential confusion.

The use of the word significant has been removed throughout.

Overall there are too many grammar/ punctuation errors which indicate the need for the article to be corrected before submission.

These have been addressed.

---

## [Decision Letter · Decision Letter 1]

21 May 2024

The impact of the COVID-19 pandemic for children in custody: an analysis of inspection reports

PONE-D-23-20933R1

Dear Dr. ennox

We’re pleased to inform you that your manuscript has been judged scientifically suitable for publication and will be formally accepted for publication once it meets all outstanding technical requirements.

Kind regards,

Massimiliano Esposito, M.D.

Academic Editor

PLOS ONE

---

## [Editor Report · Acceptance letter]

24 May 2024

PONE-D-23-20933R1 

PLOS ONE

Dear Dr. Lennox, 

I'm pleased to inform you that your manuscript has been deemed suitable for publication in PLOS ONE. Congratulations! Your manuscript is now being handed over to our production team.

Kind regards, 

on behalf of

Prof. Massimiliano Esposito 

Academic Editor

PLOS ONE